# Modernization of the Infrastructure of Marine Passenger Port Based on Synthesis of the Structure and Forecasting Development

**Srećko Krile** [1] , **Nikolai Maiorov** [2,*] **and Vladimir Fetisov** [2]

[1] Electrical Engineering and Computing Department, University of Dubrovnik, 20000 Dubrovnik, Croatia; srecko.krile@unidu.hr
[2] Department of Systems Analysis and Logistics, Saint-Petersburg State University of Aerospace Instrumentation 2, 190000 Saint Petersburg, Russia; fet1vlad@yandex.ru
[*] Correspondence: nmsoft@yandex.ru

**Abstract:** Passenger seaports are new starting-points of urban development. They form a new independent industry, become new incentives for improving urban infrastructure and increase the tourist attractiveness of the city itself and the region. In view of changes in passenger service processes, changes in route ferry and cruise networks, due to COVID-19, the heads of ports and terminals set new strategic tasks to determine the directions for infrastructure modernization and forecast development. The regions of the Adriatic and Baltic Seas were chosen as the experimental base. To find new answers, it is necessary to solve the problem of synthesizing the structure of a sea passenger port, taking into account all processes and services, the influence of the external environment, building a system of target functions and limiting conditions. Thus, the necessity of forming informed decisions on modernization based on the construction of new mathematical models is substantiated. A new function has been introduced that describes the influence of the external environment. Particular attention is given to the study of the mutual influence of the city and the sea passenger port in order to determine the need to improve transport accessibility and change the near-port transport space. The presented models of structure synthesis and target functions, models including functions of the influence of the external environment on the system "city infrastructure-sea passenger port-ferry company" allow at a qualitatively new level to solve the problem of forecasting development and form a system making decisions to improve the position of the passenger terminal in the sea region. The developed models and synthesis problem formation are applicable to sea passenger ports and terminals in other regions of the seas. The models are applicable both at the stage of creating a new marine terminal and during the study and subsequent modernization of the infrastructure. The presented new models allow the port manager to give answers to the questions of strategic development of sea passenger ports in sea regions.

**Keywords:** structure synthesis; modeling; ferry network; cruise network; mathematical model; the Baltic Sea region; sea passenger port

## 1. Introduction

Global changes in the world economy, changes in transport infrastructure, changes in society and widespread introduction of digital technologies directly affect the development and changes of ports, terminals and transport systems. Historically, there is a direct relationship between the development of ports and the development of cities. However, today there is a significant increase in the influence of such trends with competition in the market increasing environmental requirements and norms, the observed shortage of areas for ports and new terminals within urban boundaries, as well as new requirements for supply chain management and a number of others.

Until the end of the 19th century, ports were the front gates of cities and the centers of their economy and culture. Today their role has not changed either, but there has been a change in emphasis. Passenger ports are being integrated into a single transport network and should be better coordinated with river, rail, and road transport in order to ensure the highest possible speed of passenger traffic. If we consider it systematically, then the approaches to research and forecasting of the development of the seaports and terminals themselves have changed. Due to the new conditions (Covid-19), ports and terminals have faced new challenges and new difficulties. The criteria for analyzing the work of seaports, terminals and ferry companies were supplemented with conditions for ensuring the safety of passengers both in the terminal and on board.

The approaches to the study of ports from the side of society and the environment have changed, taking into account global trends [1]. The port industry with its economy nature and long payback period (different situation in regions) of port investments traditionally determines high financial needs for operating the business. However, the success of the city in tourism and the availability of jobs in the city often depends on this. This means a positive impact on economic processes and stability and that development strategies are needed. The industry expansion changed the structure, strategies and operational frameworks of passenger ports and terminals. The trend is towards bigger ships and modern diversified cruise products meeting contemporary consumer needs. Sea passenger ports strive to respond to the changing market circumstances [2].

Today, maritime cities, cities with access to the bays, are striving to adapt their coastal territories to new requirements. The main direction of the transformation of the port area is de-industrialization. In this aspect, the industrial seaport is shifted to adjacent or bulk territories, and its place is taken by cultural, commercial and tourist facilities. The port, moving to new technological spaces, is changing transport networks and infrastructure complex to ensure the convenience of passengers, but at the same time, it is not always possible to say that the transition to new territories contributes to the acceleration of the development of the port itself and some technological gain [2–4].

If we talk about the sea passenger port and ferry lines, then, of course, these type of transport systems can be quickly deployed, either by expanding the existing one, or by building a new one. Ferry services can be implemented relatively quickly with limited investments in onshore infrastructure, since the main element is an adjustable ramp (using the Ro-Ro loading and unloading method) or a berth, if tides lead to a change in the height of the embankment or berth [4–7]. This is much simpler compared to the port infrastructure of a cargo port, which requires a variety of handling equipment. With little or no handling equipment required, port operations for Ro-Ro ferries are relatively low cost. There is a slight difference if you transport only escorted trucks (with the main engine and driver) or only semi-trailers. The port operator must provide terminal transport for semi-trailers for quick loading and unloading. The following advantages of sea ferry lines are distinguished:

- by the criterion of routing organization
    - relatively fast ferry seagoing vessels;
    - short turnaround time in the port, which ensures the dynamism of the development of the entire system of cargo and passenger transportation;
    - relatively higher frequency of movement.
- according to the criterion of service in the port
    - a fairly inexpensive infrastructure is required, such as a ramp and/or a Ro-Ro berth;
    - there is practically no or not required loading and unloading equipment;
    - passengers on their own vehicles leave the ship;
    - flexibility and convenience of service for most types of cargo, as well as passengers
- issues of safety and organization of the schedule of ferry routes
    - reliable schedule due to the planning of passenger transport;
    - the risk of damage to goods is virtually excluded

The handling process in the port is very fast and efficient. In most cases, a full exchange of cargo and passengers at the seaport takes only one or two hours.

Ferry services in the regions of the seas are the main drivers of the development of interaction between cities and countries. Figure 1 shows data on the traffic intensity of ferry routes in the Baltic and Adriatic Seas for the summer navigation of 2019 with data taken from the marinetraffic.com (accessed on 20 February 2021) information system. Classification of ports in the Mediterranean and adjoining seas with respect to cruise passenger traffic (2009–2017) are detailed in the article [2].

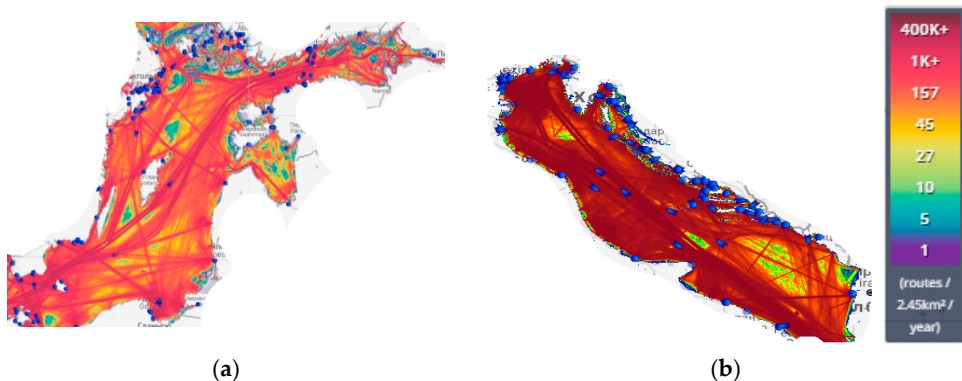

(**a**)                 (**b**)

**Figure 1.** Intensities of ferry routes in the Baltic and Adriatic Seas based on geographic information systems data in 2019–early 2020 (based on https://www.marinetraffic.com/ (accessed on 20 February 2021)): (**a**) Intensity of ferry routes in the Baltic Sea; (**b**) Intensity of ferry routes in the Adriatic Sea [7].

Despite the downturn in the industry caused by the coronavirus infection (Covid-19), there has been no change in the location and priority role of sea passenger ports and terminals. The results of studies of the theory of models and methods of development of ports and terminals since 1980 have now found their embodiment in such models as D. Byrd's model, Anyport model, UNCTAD model, and Workport model [8–11]. If we analyze them, then these models represent a step-by-step evolution of cargo seaports.

Passenger seaports remain new points of urban development, form a new independent industry, become new incentives for improving urban infrastructure and increase the attractiveness of the city itself and the region. In this aspect, the industry of sea passenger ports and terminals requires the creation of new models and methods of development, forecasting and market research. A very important task is to assess the impact of the external environment, build a functional model of the sea passenger port and, based on the data obtained, form a development strategy.

## 2. Determination of the Development Model for Sea Passenger Ports in Sea Regions

The models presented above can be applied to the development models of passenger sea ports, but their application will be limited, since the main parameters that will determine the development strategy of a passenger port are: the geographical location of the terminal; the cultural layer in the region; tourist passenger traffic and conditions for its formation; available cruise and ferry lines in the region. It is also necessary to take into account the increasing requirements for passenger mobility, the desire to travel by their own car. For the study of sea passenger ports and terminals in the Baltic Sea region the group of sea passenger terminals in St. Petersburg were selected: Passenger Port of St. Petersburg "Marine Facade", passenger terminal "Marine Vorzal". St. Petersburg also has a system of urban and commercial berths on rivers and canals, and a separate river passenger port [12,13].

The parameters of the passenger seaport can be determined by such factors: the geopolitical position of the port; geographical characteristics of the region where the port is located; the level of development of transport infrastructure and near-port transport space; the level of training of the port personnel; the nature of the processed passenger traffic;

reached stage of development of the existing route network and terminal infrastructure; strategic plans for the development of the port; the level of implementation of information technologies [14–17]. At the same time, there are the following main factors that affect the development of the sea passenger port and determine the position of the port on the market [18,19]: competition between existing ports and terminals; the impact of new entrants to the cruise product market; the impact of the external environment on society and the economic situation.

To understand port development models and highlight the proposed solution, it is necessary to investigate existing models. Among the well-known strategies for port development, the "Any Port" model stands out [12,13,19], which is a model that describes the development of a seaport in space and time. According to this model, the development of any port takes place in six main stages: the appearance and emergence of the port; expansion of the existing berths; ultimate development of the berth front; growth of the berth line; modernization of the existing berth line; specialization of berths and port areas. The author of [12] substantiated the following principles of the port development strategy: gradual growth of the port size, expansion of berths, opening of new berths; modernization and specialization of the seaport, berths, transshipment equipment related to the development of the fleet in the form of increasing the size of ships, their capacity, and the requirement to speed up the port vessel handling.

This model was applied to seaports' handling of cargo traffic and is a model describing step-by-step evolution of seaports. The model was based on the states of the ports at certain given time intervals and revealed the reasons of infrastructure growth and changes. As the model is logical enough, it has been widely used. However, due to recent changes in the economy, which have directly affected cargo flows, the model has shown insufficient versatility. The "Any Port" [20] model was built on the basis of a certain group of ports, while not enough for a more qualitative study and identification of patterns in the development of ports. On the other hand, the model did not take into account economic processes, changes in society, changes in the role of cities and, as a consequence, changes in the "megapolis city–port" relationship. However, this model laid the basis for future development models. Another model is model of UNCTAD. According to the study in [20], this model was based on the proposal to represent port development on the basis of technological aspects of port operations. The model already has five main stages of port development.

An alternative model is the "three generations" model [21]. This model has already taken into account such important sources of influence on port development as the influence and rapid growth of information systems, the introduction of electronic document management and monitoring systems for both ship traffic and cargo movement. Models of port development were reduced to the formation of three consolidated groups of generations. It is of rather general nature and does not take into account geographical location of the port, or influence of external environment on port processes. It is well known that the advantages of the geographical location of the ports determined the ways of their development and the potential for development, thus influencing the strategy and developed port facilities.

The next stage of development was the "Workport" model [21]. This model identifies eight main criteria affecting port development: form of ownership; types of cargo; organization and processes of transshipment work; information support; work culture; processes of port functions development; life safety and labor protection; ecology. The model forms the basis for system multi-criteria analysis of port development, which requires a large number of statistical data.

The models presented above can be applied to the models of development of sea passenger ports, but their application will be limited, since the main parameters that will determine the development strategy are the geographical location of the terminal, the cultural layer in the region, tourist passenger traffic and the conditions of its formation.

Among the sources of literature on the models of port development, one can note both the limited presentation of information and the geographical locality. Characteristic

examples are the sources [10–12], which present the peculiarities of geographical location of the port, and local milestones of development only for the respective ports.

On the basis of a retrospective analysis of changes in the passenger ports of St. Petersburg, a new model of interaction "city (urban transport systems)—sea passenger port" was built. This model introduces the boundaries of the near-terminal transport space of the seaport, the historical part of the city as a center of attraction for tourists, and the boundaries of the metropolis itself.

Figure 2 shows the backbone interaction with the historical center of the city, with large passenger terminals on other modes of transport (directions are highlighted in bold lines), and interaction with the existing urban route network. It should be noted that each stage of the development of a sea passenger port is characterized by a corresponding level of development of the adjacent urban transport infrastructure and the presence of its own branched route network. The implementation of all the presented routes increases the attractiveness of the sea passenger port for tourists, improves its accessibility and increases the attractiveness of the city. The tourist independently forms his route, in accordance with the targets. For example, he can choose to walk in a new area near the sea passenger port and not go to the historic center. As an alternative, he will reach the center on his own by public transport. The presence of a new microdistrict near the port allows the development of new activities, and the formation of new centers of attraction, which contributes to the development of passenger mobility, and this equally contributes to the new development of the city, as does the presence of a passenger port in the historically established proximity to the center.

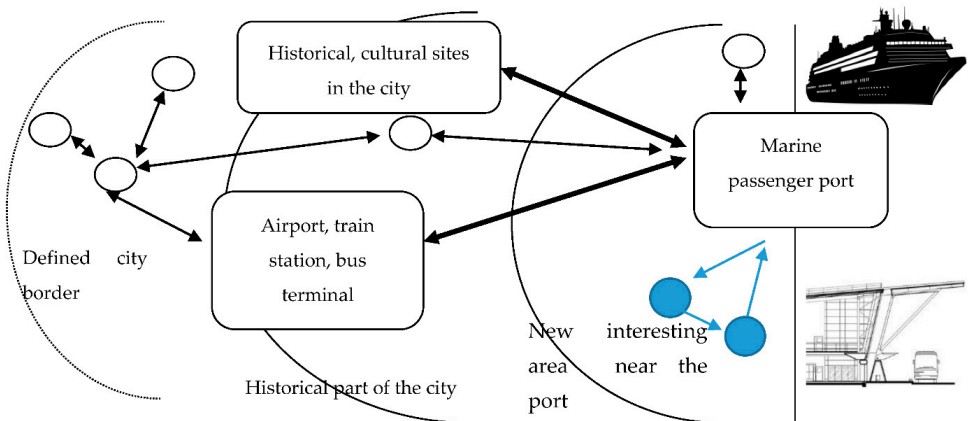

**Figure 2.** Model of the mutual influence of the city and the sea passenger port.

In the context of the restoration of passenger traffic after the negative impact of Covid-19 on this industry, there will be changes both in the route network and in the processes of passenger service.

Figure 3 shows the three stages of change in system marine passenger port and ferry routes in region of sea. The first stage looks at the situation before Covid-19. There is an already existing and stable ferry route network in the sea regions. The second stage is to stop the ferry service, closing borders due to the Covid-19 pandemic. At the second stage, marine passenger ports and terminals should improve infrastructure and services, and improve the processes of servicing future passengers. It is at the third stage that the gradual restoration of ferry communication takes place. However, at the same time, in the regions of the seas, the positions of ports and terminals relative to each other in the passenger transportation market can be changed. Passenger ports and terminals offering and providing the best service and security will become the leaders in the regions. If we implement the model (Figure 4) on the basis of set theory, then we can form the following scenarios in the strategy of the marine passenger port:

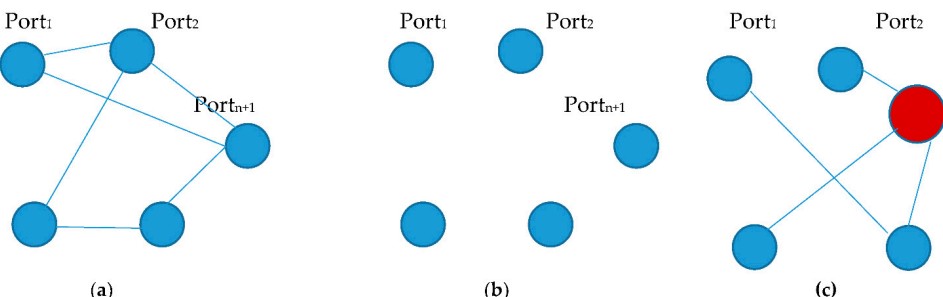

**Figure 3.** Model of changes in the route network of ferry lines after restoration and opening of borders: (**a**) Ferry system in 2019; (**b**) Ferry system in 2020; (**c**) Ferry after restart in 2021.

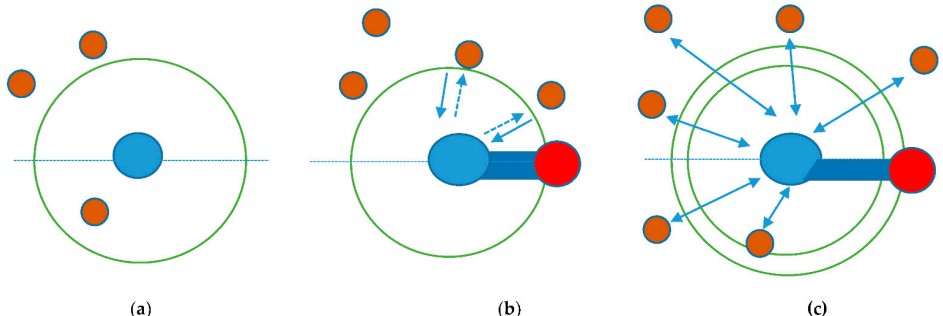

**Figure 4.** Model of changes in the route network of ferry lines after restoration and opening of borders: (**a**) Model of separate sea passenger port; (**b**) The port changes the near-port space, while there is interaction with one other sea passenger port; (**c**) The sea passenger port increases its market share. In Figure 4, ⬤—new passenger ports; ⬤—original sea passenger port; ◯—port area of interest; ⬌—ferry routes between ports; ⬤—regular routes between sea passenger ports (e.g., Helsinki–Tallinn in Baltic sea).

## 3. Mathematical Model of the Sea Passenger Port System and the Formation of Alternatives for the Choice of Infrastructure Modernization

The task of synthesizing structures is difficult to formalize, due to the requirement to take into account a large number of system parameters. The system of a passenger sea terminal, as a rule, has the following organizational and functional structure:

1. subsystem of organizational support, subsystem for service cruise and ferry ships and sea berths;
2. a subsystem of information support, which is an organizing link in the work of the system and performing information exchange;
3. marine passenger terminal, which is a production unit, a place for performing all technological operations with passenger traffic;
4. subsystem of service support, work planning and support of ship and passenger traffic support.

The objective function of optimizing the structure and operation of the sea passenger terminal is multi-parameter and multi-criteria. When implementing it, it is necessary to take into account the number of passenger terminals, berths, and the intensity of ships' calls [22,23]. The target function is representable by Equation (1):

$$T(I, W, S, G, U) = [F_1(I, W, S, G, U) \rightarrow \max, \ldots, F_m(I, W, S, G, U) \rightarrow \max], \quad (1)$$

where $T$—vector of criteria for assessing the efficiency of the passenger seaport; $F_1, \ldots, F_m$—performance criteria for individual port terminals and berths; $I$—vector of parameters

describing the structure of the incoming flow of cruise ships and additional transport; *W*—vector of parameters describing the structure of the outgoing flow of transport and ferry ships; *S*—vector of parameters characterizing the technical equipment of terminals and berths; *G*—vector of parameters describing the work schedule and organization of processing of passenger traffic and associated cargo traffic; *U*—vector of parameters characterizing the services and services provided by the terminals.

The system of the sea passenger terminal is formalized by a mathematical model of the form:

$$\sum = \left(T, \overline{X}, \Omega, \overline{Y}, \Gamma, G, Z, H, F, E_0\right), \tag{2}$$

where $\sum$—marine passenger terminal system; $T = \{(t_i, t_{i+1})\}_1^N$—time, intervals of terminal activity; $\overline{X} = \{x_j\}_1^N$—input information (passenger traffic; ferry calls; working hours of services and port departments); $\Omega = \{\omega_i\}_1^N, \omega_i \in \Omega$—input operator, set $\Omega$—input actions; $\overline{Y} = \{y_i\}_1^M$—results (the process of processing passengers, cargo traffic and ships); $\Gamma = \{\gamma_i\}_1^M$—operator for outputting results to the external environment (operator reflecting the results of the port operation); $G = \{g_i\}_1^k$—output function (input to output conversion algorithm); $Z = \{z_i\}_1^k$—set of internal states of the passenger terminal system; $H = \{h_i\}_1^I$—transition function (algorithm, process of using internal resources); $F = \{f_i\}_1^I$—process control function of the sea passenger port; $E_0 = \{e_i\}_1^L$—aftereffect function (result of a previous system action or system memory in the form of discrete states of port development levels).

For the stable functioning of such a system and conducting a comprehensive analysis, the following conditions must be met:

- The passenger sea terminal must have executive and control elements. Executive elements participate in the transformation of the input into the output; control elements do not transform, but act on elements—converters;
- They have an entrance and an exit ($\overline{X}$ and $\overline{Y}$; passenger flow and ship flow), which connect the system with the external environment, determine the type of system and set the characteristics of the processes;
- They have a control function *F*, the purpose of which is to influence the entire system as a whole. In addition, the control function must have the property of adaptation to changing environmental conditions;
- They have a target *F* (target function), the achievement of which is regulated by a regulating device R that implements the control function *F*;
- They have a regulating device R, which controls the operation of the system through feedback, acting on the system through feedback. In this case, it is necessary to have blocks for collecting and analyzing work on servicing passengers and ships, a department that analyzes operational work;
- They have availability of a function that uses both external and internal *Z* resources of the system—(*H*);
- They have the presence of feedback between *G*, *H*, *Z*, *E*.

In the case of the situation presented in Figures 2–4, port leaders need to implement solutions to improve services and improve passenger handling processes. This situation requires solving the problem of forecasting development, which is determined by various modes of operation of the port system. It should be noted that the new solutions are based on infrastructure in 2019 and the beginning of 2020. Now, it is necessary to create various scenarios.

The variant of organizing the work of the seaport system can be formed only when focused on the function that each element of the system must provide. Taking into account the principles of constructing the system of tasks in the future, and the set of existing and future objects of influence, the system of functions should reflect the full set of functions of the "ideal" system "passenger seaport—ferry company". The implementation of this set will make it possible to form the appearance of the forecast object, in our case, the

sea passenger port and route network. As a result of this analysis, in the first case, the alternatives of the appearance of the marine system are determined. When specifying various concepts of the seaport system, the system of functions can change in the direction of expanding the functions performed:

$$\left\{\varphi^{main}\right\} \rightarrow \left\{\varphi^1\right\} \cap \left\{\varphi^2\right\} \cap \left\{\varphi^3\right\} \cap \ldots \left\{\varphi^N\right\} \qquad (3)$$

where *N*—number of options for organizations of the maritime port system.

As a result of the approximations of the functions, the model of the system is constructed based on the expression:

$$\left\{\varphi^I\right\} \rightarrow \left\{\varphi^{main}\right\} \pm \Delta\varphi^{forecast}, \qquad (4)$$

To perform actions, you must enter the graph of the funds system $S = \left\{S^0, S^1, S^2, \ldots, S^{n-1}\right\}$ which ensures the performance of functions at every level. This type of funds $S_j \in \left\{S^{\eta}_{1j}\right\} \varphi$ provides the fundamental possibility of performing the function $\varphi^{\eta}_{1i} \in \varphi^0$ ($\eta$—hierarchy level of the system presentation).

Then, the problem of synthesizing alternatives for constructing marine passenger port system consists in a directed choice from the set $\left\{S^0\right\}$ of elements that provides a set of elements of a system of functions for different variants $\left\{\varphi^0\right\}$. We denote the synthesis model by $A_\varphi$. Model for the synthesis of alternatives to the appearance of the system $A_\varphi$ are given by the set of level mappings of the system $S^0$ in system.

Alternative options for organizing the system of a sea passenger port can be formulated as follows: it is necessary to develop principles, approaches, models, methods to find the most preferable options for organizing and predicting the development of a sea passenger port based on the model:

$$A_\psi : \left[\left\{\varphi^0_\psi\right\} \rightarrow \left\{S^0_\psi\right\}; \left\{\varphi^1_\psi\right\} \rightarrow \left\{S^1_\psi\right\}, \ldots, \left\{\varphi^N_\psi\right\} \rightarrow \left\{S^N_\psi\right\}\right], \qquad (5)$$

With restrictions on the parameters of the functional efficiency of system functions:

$$\overline{P}^1_{f_{\min}} \leq \overline{P}^1_j \leq \overline{P}^1_{f\max}; P^1_f = \left\{P^1_1; P^1_2; P^1_3, \ldots, P^1_{K_1}\right\}$$
$$\ldots\ldots\ldots\ldots\ldots\ldots\ldots\ldots\ldots\ldots\ldots\ldots\ldots \qquad (6)$$
$$\overline{P}^N_{f_{\min}} \leq \overline{P}^N_j \leq \overline{P}^N_{f\max}; P^N_f = \left\{P^N_1; P^N_2; P^N_3, \ldots, P^N_{K_1}\right\}$$

Given the connections, the sets $\varphi$ and $S$:

$$\left\{\varphi^0\right\} \times \left\{S^0\right\} \rightarrow \left\{\theta^0\right\}$$
$$\ldots\ldots\ldots\ldots\ldots \qquad (7)$$
$$\left\{\varphi^N\right\} \times \left\{S^N\right\} \rightarrow \left\{\theta^N\right\}$$

Given the connections, the sets $\varphi$ and $\{F, \nu\}$:

$$\left\{\varphi^1\right\} \times \left\{\varphi^2\right\} \rightarrow \left\{F^2, \nu^{1-2}\right\}$$
$$\ldots\ldots\ldots\ldots$$
$$\left\{\varphi^{N-1}\right\} \times \left\{\varphi^N\right\} \rightarrow \left\{F^N, \nu^{N-1,N}\right\},$$
$$\left\{S^1\right\} \times \left\{S^2\right\} \rightarrow \left\{\varepsilon^{1-2}\right\} \qquad (8)$$
$$\ldots\ldots\ldots\ldots\ldots\ldots$$
$$\left\{S^N\right\} \times \left\{S^{N-1}\right\} \rightarrow \left\{\varepsilon^{N,N-1}\right\}$$

where $\psi$ is the index of the alternative to the organization of the system $0 \leq \psi \leq M$; *M*—a limited number of alternatives for organizing the work of systems; $\eta$—the level of the hierarchy of the system presentation; *N* is the number of levels in the hierarchy; $\overline{P}^1_{f_{\min}}$,

$\overline{P}^1_{f_{\max}}$—the range of parameters of the functioning of efficiency relative to the $\eta$-th level of the hierarchy; $k$ is the number of parameters of functional efficiency at the $\eta$-th level; $\theta$—communication of $\varphi$ and $S$; $F$—link between the levels of the hierarchy of the system $\varphi$; $\nu$—communication within the level system $\varphi$; $\varepsilon$—communication within $S$.

The appearance of the system is set in the form of a set of hierarchically organized sets of functional subsystems and elements. Parameters of functional efficiency are set relative to the functions derived for each level of decomposition. The method of enumeration within a set $S^N$ by connections forms a set of alternative variants of organization (forms) at the elementary level $A^N_\psi$. Considering these sets under the conditions of parameter constraints, alternatives are selected $A^N_\psi(P^N_f) \in A_\psi{}^N$ that provide the specified ranges of parameter variation.

On the basis of the presented approach and the formation of the problem of synthesizing the structure of a marine passenger port, it is possible to form a system of target functions:

$$
\begin{cases}
\sum\limits_{k=1}^{c} l_k T_k - \sum\limits_{i=1}^{p} L_{k1} T_{ik}(S,G) \to \min \\
C_i n_i(S,G) + \sum\limits_{j=1}^{h} (C_{j1} + C_j W_j) + \sum\limits_{k=1}^{c} C_k L_k^1 T_k(S,G) + C_k L_k^2 T_{k1}(S,G) \to \min \\
F_i(\alpha_i) \to \min \\
\sum\limits_{j=1}^{h} (K_j^{\max}(G,S) - K_j(G,S)) \to \min
\end{cases}
, \quad (9)
$$

under the following model constraints:
$$
\begin{cases}
0 \le K_j^{\max} \le 1, j \in 1..n \\
a_i \le \alpha \le b_i, \alpha \in 1..n, b \in 1..n \\
T_{kj}^{\max}(G,S) \le T_{kj}^{\max}, j \in 1..p, k \in 1..c
\end{cases}
$$

where $p$, $c$—the number of technological stages for processing passenger traffic, the number of passenger requests in the system; $h$—the number of resources of passenger terminals (areas and volumes of terminals, berths); $C_i$—possible delays in handling passengers or ships; $n_i(S,G)$—the number of ships that received delays in a certain time interval; $C_{j1}$—the cost of operating the $j$-th terminal resource during the considered time; $C_j$—operating cost per unit area of the sea terminal; $W_j$—area occupied by the $j$-th resource; $C_k$—some penalty for waiting for a vessel to be processed or possible terminal downtime; $L_k^1$—indication of processing of a specific $k$-th ferry or cruise ship; $T_k(S,G)$—duration of unloading of the $k$-th ship; $l_k$—importance of the processing time indicator for the $k$-th order for the terminal; $T_{k1}(S,G)$—duration of loading the $k$-th application; $L_k^2$—sign of loading the $k$-th order; $T_k$—required duration of processing of vessels; $L_{ki}$—sign of the passage of the $i$-th stage by the $k$-th order in the port; $T_{ik}(S,G)$—duration of the $i$-th stage of processing the $k$-th order; $K_j(G,S)$—loading the $j$-th terminal resource; $K_j^{\max}$—maximum possible required loading of the $j$-th port resource; $T_{kj}^{\max}$—maximum possible duration of order processing at the $j$-th stage of technological operations; $F_i(\alpha_i)$—environmental influence function; $a_i$, $b_i$—boundaries of change, determined by the researcher or external constraints.

The inclusion of a variable $F_i(\alpha_i)$ is necessary to take into account the influence of the external environment. The striving of this variable to a minimum means the striving of the system to achieve stable operation of the marine passenger port terminal and the absence of competition in the ferry market between different terminals. There is a choice of equipment for the terminal between those available on the market when synthesizing the structure of the terminal. Each type of equipment has its own performance, cost to build, cost to buy or rent and operate with logistics and equipment refurbishment. Consequently, at the stage of determining the type and amount of resources used, the parameters of the synthesis problem are discrete. It should be noted that functions such as $K_j(G,S)$, $T_{ik}(S,G)$ and others reflecting the duration of work, and loading of elements are non-linear functions,

therefore, when investigating them, it is desirable to introduce a division into certain time intervals.

## 4. Discussion and Conclusions

Without constructing the objective function, it is impossible to construct a high-quality modeling, since in this case, all the required parameters will not be taken into account. Otherwise, the marine passenger terminal model will be very generalized and imprecise. Practical implementation in the form of a simulation model will make it possible to refine the results obtained, determine the spatial arrangement of equipment and use real values in the model. When setting the objective function, the areas of feasible decisions are immediately determined, which speeds up the process of subsequent analysis and decision-making.

The obtained objective functions allow setting and solving an optimization problem for the synthesis of the structure of a sea passenger terminal. This task is extremely important for conducting a study of the terminal system as a whole, taking into account the peculiarities of the interaction between the elements and solving the forecast problem. Based on the proposed methodology and practical implementation, it is possible to form a reasoned choice of the best organization of the terminal operation. From a scientific standpoint, the construction of such functions is an important research result. The synthesis is carried out in order to substantiate the set of elements of the port structure, relationships and connections between them, characteristics of the elements and connections that together provide the best functioning and perform the subsequent research through modeling. However, the obtained dependences show that when performing calculations, the given systems of equations will require more detail and the inclusion of each functional of the terminal operation, taking into account the boundary conditions. The inclusion of the function describing the influence of the external environment allows, when implementing in the simulation model, to take into account the stochastic nature of the processes and is one of the further areas of research in the considered operating conditions of sea passenger terminals. The model is built on the basis of the analytical form of the objective function, decomposition of the terminal structure and substantiation of the implementation conditions for each section, and further research aimed at achieving the best functioning of the marine passenger terminal system is advisable to be carried out based on the methods of simulation and the theory of similarity of systems.

The presented mathematical models take into account the influence of the external environment, which is extremely important in the situation with the assessment of the sphere of sea passenger traffic due to COVID-19. The above equations allow us to study, in a new way, the issues of modernizing the infrastructure of the sea passenger port, to identify new weaknesses in the processing of passenger flows based on new safety requirements. The solutions presented make it possible to improve both the decision-making system for the modernization of the sea passenger port infrastructure, and the reliability of the "sea passenger port-ferry company" system based on the construction of equations for the synthesis of systems.

**Author Contributions:** Data collection and analysis, concept of the research, S.K.; Mathematical models, writing an article, text edition for the final version, drafting literature review, critical revision of the article, supervisions, N.M., V.F. All authors have read and agreed to the published version of the manuscript.

**Funding:** This research received no external funding.

**Institutional Review Board Statement:** Not applicable.

**Informed Consent Statement:** Not applicable.

**Data Availability Statement:** Not applicable.

**Conflicts of Interest:** The authors declare no conflict of interest.

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
