# Peer review of "Modernization of the Infrastructure of Marine Passenger Port Based on Synthesis of the Structure and Forecasting Development"

_sustainability, doi:10.3390/su13073869_

Round 1
Reviewer 1 Report
- English language is not up to a standard, moderate changes required (one example: “For the study of sea passenger ports and terminals in the Baltic Sea region were selected the group of sea passenger terminals in St. Petersburg!”)
- Plagiarism check (computerized) showed 12% (cumulative) which is more than OK.
- Only questionable item noted is Figure 1. It is copied from article published by same authors, without referencing (reflist, item 7). As there are no other issues of this kind, this is probably unintentional mistake.
- Figure 1 is hard to understand, it should be explained (also, correct Baltic SeaL)
- There are some formatting errors
- Explanation of the Figure 4 is dislocated, should be fixed,
- Chapter 3 should start on the next page,
- Size of variables in text should be corrected (at the moment they are too large),
- Equations number must be aligned,
- Minor mistakes in the references,
- Equations sizes to be adjusted acc. to the journal recommendations,
- Author Biography is missing in the article, according to the “Instructions for Authors“, it should be a part of the article (it is strong recommendation, not mandatory).
- Just one recommendation, non-mandatory improvement: Abstract is slightly misleading, at least for non-native speaker. It took me few attempts to understand that the model is applicable across the board, not only in Adriatic and Baltic. That should be clearly stated in the abstract.
Author Response
Response to Reviewer 1 Comments
Point 1: English language is not up to a standard, moderate changes required (one example: “For the study of sea passenger ports and terminals in the Baltic Sea region were selected the group of sea passenger terminals in St. Petersburg!”) 

Response 1: Please provide your response for Point 1. (in red)
A correction and clarification in the text has been made.
For the study of sea passenger ports and terminals in the Baltic Sea region were selected such terminals: Passenger Port of St. Petersburg "Marine Facade", passenger terminal "Marine Vorzal".
Point 2: Plagiarism check (computerized) showed 12% (cumulative) which is more than OK.
Response 2: Please provide your response for Point 2. (in red)
Research in this area has been conducted by the authors for a long time. This article presents the resulting models and methods.
Point 3: Only questionable item noted is Figure 1. It is copied from article published by same authors, without referencing (reflist, item 7). As there are no other issues of this kind, this is probably unintentional mistake.
Response 3: Please provide your response for Point 2. (in red)
A reference to the authors' article has been added in the caption to the figure 1. A similar graph is presented in the article [7] in the list of references
Point 4: Figure 1 is hard to understand, it should be explained (also, correct Baltic SeaL)
Response 4: Please provide your response for Point 2. (in red)
An error in the text has been corrected. The correction is highlighted in red text.
Point 5: There are some formatting errors
- Explanation of the Figure 4 is dislocated, should be fixed,
- Chapter 3 should start on the next page,
- Size of variables in text should be corrected (at the moment they are too large),
- Equations number must be aligned,
- Minor mistakes in the references,
- Equations sizes to be adjusted acc. to the journal recommendations,
Response 5: Please provide your response for Point 2. (in red)
An error in the text has been corrected. The correction is highlighted in red text.
- Corrections have been made, and the pictures have been grouped together.
- Chapter 3 moved to the new page
3-6. Adjustments have been made to the article
Point 6: Author Biography is missing in the article, according to the “Instructions for Authors“, it should be a part of the article (it is strong recommendation, not mandatory).
Response 6: Please provide your response for Point 2. (in red)
Information about the authors was added when the article was registered
Point 7: Just one recommendation, non-mandatory improvement: Abstract is slightly misleading, at least for non-native speaker. It took me few attempts to understand that the model is applicable across the board, not only in Adriatic and Baltic. That should be clearly stated in the abstract.
Response 7: Please provide your response for Point 2. (in red)
Passenger seaports are new starting-points of urban development. They form a new independent industry; become new incentives for improving urban infrastructure and increasing the tourist attractiveness of the city itself and the region. In view of changes in passenger service processes, changes in route ferry and cruise networks, due to Covid-19, the heads of ports and terminals set new strategic tasks to determine the directions for infrastructure modernization and forecast development. The regions of the Adriatic and Baltic Seas were chosen as the experimental base. To find new answers, it is necessary to solve the problem of synthesizing the structure of a sea passenger port, taking into account all processes and services, the influence of the external environment, building a system of target functions and limiting conditions. Thus, the necessity of forming informed decisions on modernization based on the construction of new mathematical models is substantiated. A new function has been introduced that describes the influence of the external environment. Particular attention is given to the study of the mutual influence of the city and the sea passenger port in order to determine the need to improve transport accessibility and change the near-port transport space. The presented models of structure synthesis and target functions, models including functions of the influence of the external environment on the system "city infrastructure-sea passenger port-ferry company" allow at a qualitatively new level to solve the problem of forecasting development and form a system making decisions to improve the position of the passenger terminal in the sea region. The developed models and synthesis problem formation are applicable to sea passenger ports and terminals in other regions of the seas. The models are applicable both at the stage of creating a new marine terminal and during the study and subsequent modernization of the infrastructure.

Reviewer 2 Report
Authors must make the following corrections to the article:
- Abstract needs to be rebuilt. The authors must precisely define the purpose of the research in this section. It would be appropriate to quantify the results obtained.
- Authors should explain the academic contribution of the work prepared. The table of authors' contributions is essential so that the reader can understand the novelty of his work compared to existing literature.
- There is no chapter of "Literature Review". Without the comparative analysis of the previous papers, the models and methodological approach of the authors are not convincible or reliable at all. Make a separate chapter for the background of the model and methodologies, so that the novelty of the research conducted can be properly understood.
- Managerial input and a case study are required to show the actual application of the model. Statistics are missing.
- The authors should reconstruct the conclusions from the work and refer to the subsequent stages of the work in more detail.
Author Response
Response to Reviewer 2 Comments
Point 1: Abstract needs to be rebuilt. The authors must precisely define the purpose of the research in this section. It would be appropriate to quantify the results obtained.

Response 1: Please provide your response for Point 1. (in red)
Corrections have been made to the abstract.
Regarding the numerical parameters for comparing the application of the methodology, at this time, research and infrastructure changes are made in the selected ports. The results will be obtained on the basis of the analysis of the new navigation, when the restrictions will be removed. A number of ports did not change anything, did not change the infrastructure. Others, on the contrary, used the time to improve services and infrastructure.
Passenger seaports are new starting-points of urban development. They form a new independent industry; become new incentives for improving urban infrastructure and increasing the tourist attractiveness of the city itself and the region. In view of changes in passenger service processes, changes in route ferry and cruise networks, due to Covid-19, the heads of ports and terminals set new strategic tasks to determine the directions for infrastructure modernization and forecast development. The regions of the Adriatic and Baltic Seas were chosen as the experimental base. To find new answers, it is necessary to solve the problem of synthesizing the structure of a sea passenger port, taking into account all processes and services, the influence of the external environment, building a system of target functions and limiting conditions. Thus, the necessity of forming informed decisions on modernization based on the construction of new mathematical models is substantiated. A new function has been introduced that describes the influence of the external environment. Particular attention is given to the study of the mutual influence of the city and the sea passenger port in order to determine the need to improve transport accessibility and change the near-port transport space. The presented models of structure synthesis and target functions, models including functions of the influence of the external environment on the system "city infrastructure-sea passenger port-ferry company" allow at a qualitatively new level to solve the problem of forecasting development and form a system making decisions to improve the position of the passenger terminal in the sea region. The developed models and synthesis problem formation are applicable to sea passenger ports and terminals in other regions of the seas. The models are applicable both at the stage of creating a new marine terminal and during the study and subsequent modernization of the infrastructure. The presented new models allow the port manager to give answers to the questions of strategic development of sea passenger ports in sea regions.
Point 2: Authors should explain the academic contribution of the work prepared. The table of authors' contributions is essential so that the reader can understand the novelty of his work compared to existing literature.
.

Response 2: Please provide your response for Point 1. (in red)
According to observation 3 the text placed in the second chapter is prepared.
Point 3: There is no chapter of "Literature Review". Without the comparative analysis of the previous papers, the models and methodological approach of the authors are not convincible or reliable at all. Make a separate chapter for the background of the model and methodologies, so that the novelty of the research conducted can be properly understood.
.

Response 3: Please provide your response for Point 1. (in red)
According to observation 3 the text placed in the second chapter is prepared. New books and analytical reports have been added, which are based on modern models of port and terminal development. These models do not take into account the influence of the external environment and are very limited in their practical application. The peculiarity of the presented approach is the formation of the synthesis problem taking into account the influence of the external environment.
To understand port development models and highlight the proposed solution, it is necessary to investigate existing models. Among the well-known strategies for port development, the "Any Port" model stands out [12,13,19]. A model that describes the development of a seaport in space and time. According to this model, the development of any port takes place in six main stages: the appearance and emergence of the port; expansion of the existing berths; ultimate development of the berth front; growth of the berth line; modernization of the existing berth line; specialization of berths and port areas. The author of [12] substantiated the following principles of the port development strategy: gradual growth of the port size, expansion of berths, opening of new berths; modernization and specialization of the seaport, berths, transshipment equipment, related to the development of the fleet in the form of increasing the size of ships, their capacity, the requirement to speed up the port vessel handling.
This model was applied to seaports handling cargo traffic and is a model describing step-by-step evolution of seaports. The model was based on the states of the ports at certain given time intervals and revealed the reasons of infrastructure growth and changes. Because the model is logical enough, it has been widely used. However, due to recent changes in the economy, which have directly affected cargo flows, the model has shown insufficient versatility. The “Any Port” [20] model was built on the basis of a certain group of ports, while not enough for a more qualitative study and identification of patterns in the development of ports. On the other hand, the model did not take into account economic processes, changes in society, changes in the role of cities and, as a consequence, changes in the " city - metropolis - port" relationship. However, this model laid the basis for future development models. Another model is model of UNCTAD. According to [20] this model was based on the proposal to represent port development on the basis of technological aspects of port operations. The model already has five main stages of port development.
An alternative model is the "three generations" model [21]. This model has already taken into account such important sources of influence on port development as the influence and rapid growth of information systems, the introduction of electronic document management and monitoring systems for both ship traffic and cargo movement. Models of port development were reduced to the formation of three consolidated groups of generations. It is of rather general nature and does not take into account geographical location of the port, influence of external environment on port processes. It is well known that the advantages of the geographical location of the ports determined the ways of their development and the potential for development, thus influencing the strategy and developed port facilities.
The next stage of development was the "Workport" model [21]. This model identifies eight main criteria affecting port development: form of ownership; types of cargo; organization and processes of transshipment work; information support; work culture; processes of port functions development; life safety and labor protection; ecology. The model forms the basis for system multicriteria analysis of port development, which requires a large number of statistical data.
The models presented above can be applied to the models of development of sea passenger ports, but their application will be limited, since the main parameters that will determine the development strategy: the geographical location of the terminal, the cultural layer in the region, tourist passenger traffic and the conditions of its formation.
Among the sources of literature on the models of port development one can note both the limited presentation of information and the geographical locality. Characteristic examples are the sources [10,11,12], which present the peculiarities of geographical location of the port, local milestones of development only for the respective ports.
Point 4: Managerial input and a case study are required to show the actual application of the model. Statistics are missing.
.
.

Response 4: Please provide your response for Point 1. (in red)
The presented mathematical model is constructed on the basis of the data on operation of ports and terminals before restrictions in connection with COVID-19. Data on port operations and ferry companies are publicly available.
The article presents the formation of the problem of structure synthesis, taking into account the influence of the external environment. The presented solution and model allow solving the problem of modeling different scenarios of infrastructure modernization, solve the problem of strategic development.
Point 5: The authors should reconstruct the conclusions from the work and refer to the subsequent stages of the work in more detail.
..

Response 5: Please provide your response for Point 1. (in red)
The correction of the conclusion is done.
Without constructing the objective function, it is impossible to construct a high-quality modeling, since in this case all the required parameters will not be taken into account. Otherwise, the marine passenger terminal model will be very generalized and imprecise. Practical implementation in the form of a simulation model will make it possible to refine the results obtained, determine the spatial arrangement of equipment and use real values in the model. When setting the objective function, the areas of feasible decisions are immediately determined, which speeds up the process of subsequent analysis and decision-making.
The obtained objective functions allow setting and solving an optimization problem for the synthesis of the structure of a sea passenger terminal. This task is extremely important for conducting a study of the terminal system as a whole, taking into account the peculiarities of the interaction between the elements and solving the forecast problem. Based on the proposed methodology and practical implementation, it is possible to form a reasoned choice of the best organization of the terminal operation. From a scientific standpoint, the construction of such functions is an important research result. The synthesis is carried out in order to substantiate the set of elements of the port structure, relationships and connections between them, characteristics of the elements and connections that together provide the best functioning and perform the subsequent research through modeling. However, the obtained dependences show that when performing calculations, the given systems of equations will require more detail and the inclusion of each functional of the terminal operation, taking into account the boundary conditions. The inclusion of the function describing the influence of the external environment allows, when implementing in the simulation model, to take into account the stochastic nature of the processes and is one of the further areas of research in the considered operating conditions of sea passenger terminals. The model is built on the basis of the analytical form of the objective function, decomposition of the terminal structure and substantiation of the implementation conditions for each section, and further research aimed at achieving the best functioning of the marine passenger terminal system is advisable to be carried out based on the methods of simulation and the theory of similarity of systems.
The presented mathematical models take into account the influence of the external environment, which is extremely important in the situation with the assessment of the sphere of sea passenger traffic due to COVID-19. The above equations allow us to study in a new way the issues of modernizing the infrastructure of the sea passenger port, to identify new weaknesses in the processing of passenger flows based on new safety requirements. The solutions presented make it possible to improve both the decision-making system for the modernization of the sea passenger port infrastructure, and the reliability of the “sea passenger port-ferry company” system based on the construction of equations for the synthesis of systems.

Reviewer 3 Report
The topic is properly presented and the ideas of the authors are clear.
It would be interesting to provide further analysis with respect to the data to be collected once the pandemic crisis is over.
A couple of small adjustments should be done with respect to fig. 1:
the colorbars are missing and there is a type in the caption (Baltic Seal -> Baltic Sea).
Author Response
Response to Reviewer 3 Comments
Point 1: A couple of small adjustments should be done with respect to fig. 1:
the colorbars are missing and there is a type in the caption (Baltic Seal -> Baltic Sea).
Response 1: Please provide your response for Point 1. (in red)
A colorbars has been added to figure 1 to show the intensity data (ferry and cruises lines in the region of seas) and the normalization value of this colorbars. Corrections have been made to the title of figure 1.
The graphs for the Baltic Sea and Adriatic Sea region have been improved in the article.

Round 2
Reviewer 2 Report
Artykuł może zostać opublikowany w dotychczasowej formie.